# MULTIMODAL ATTRIBUTED GRAPHS: BENCHMARKING AND RETHINKING

## ABSTRACT

Associating unstructured data with structured information is crucial for real-world tasks that require relevance search. However, existing graph learning benchmarks often overlook the rich semantic information associated with each node, ignoring other available modalities such as the corresponding images. To bridge this gap, we introduce the Multimodal Graph Benchmark (MM-GRAPH), the first comprehensive multi-modal graph benchmark that incorporates both textual and visual information, going beyond the prior focus on just text-attributed graphs. MM-GRAPH consists of seven graph learning datasets of various scales that are appropriate for different learning tasks, and enable a comprehensive evaluation of graph learning algorithms in real-world scenarios thanks to their multimodal node features. To facilitate research on multimodal graph learning, we further provide an extensive study on the performance of various graph learning frameworks in the presence of features from various modalities. MM-GRAPH aims to foster research on multimodal attributed graphs and drive the development of more advanced and robust multimodal attributed graph learning algorithms. By providing a diverse set of datasets and benchmarks, MM-GRAPH enables researchers to evaluate and compare their models in realistic settings, ultimately leading to improved performance on real-world applications that rely on multimodal attributed graphs.

## 1 INTRODUCTION

Graphs are ubiquitous data structures that can effectively represent complex relationships and interactions between entities in various domains, such as social networks, biological systems, and recommendation systems. In real-world scenarios, these entities often possess rich semantic information in the form of unstructured data, such as images and text descriptions. Associating these unstructured data with the structured graph information is essential for tasks that require relevance search and information retrieval, such as recommendation systems Su et al. (2021); Crandall et al. (2009); Guo et al. (2011).

Despite the importance of multimodal graph learning, existing graph learning benchmarks typically focus on investigating graphs with various connectivity patterns, such as long-range links, out-of-distribution scenarios, etc Dwivedi et al. (2022); Gui et al. (2022); Li et al. (2024b); Morris et al. (2020); Hu et al. (2020). These benchmarks aim to evaluate the performance of graph neural networks (GNNs) and other graph learning algorithms on different graph structures and topologies. However, these benchmarks under-utilize the rich semantic information naturally present in each node, limiting their ability to evaluate graph learning algorithms in realistic settings.

With the prevalence of language modeling, there has been an outbreak in the effective integration of text and graph topology on text-attributed graphs, leading to a comprehensive benchmark on text-attributed graphs Yan et al. (2023); Zhu et al. (2024a); Jin et al. (2023a); Peng et al. (2024). While textual information undoubtedly plays a crucial role in understanding the data and making predictions for user-centric tasks, it is essential to recognize that other modalities, particularly visual information, can be helpful and provide complementary information that text alone cannot capture. For example, as shown in Figure 1(a), visually similar products may have distinct text features and ignoring such semantic gap between modalities may prevent GNNs from fully exploiting the relationships between multimodal semantics and structures.

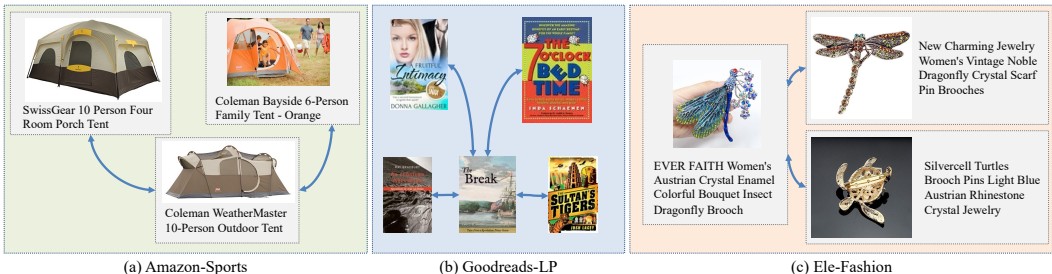

(a) Amazon-Sports        (b) Goodreads-LP        (c) Ele-Fashion

Figure 1: Visualization of our Multimodal Graph Benchmark. All nodes of our benchmark have both visual and text features. **(a) Amazon-Sports:** The image and text come from the original image and title of the sports equipment. **(b) Goodreads-LP:** The image comes from the cover of the book. We do not show the text features of Goodreads-LP since the book description is very long. **(c) Ele-fashion:** The image and text come from the original image and title of the fashion product.

To address this limitation, we introduce the Multimodal Graph Benchmark (MM-GRAPH), the first comprehensive graph benchmark that incorporates both textual and visual information. MM-GRAPH consists of seven graph learning datasets of various scales, designed for diverse downstream tasks such as node classification, link prediction and knowledge graph completion. These datasets contain multimodal node features, including text and images, alongside the traditional graph structure. By providing a diverse set of datasets and benchmarks, MM-GRAPH enables researchers to rigoriously evaluate and compare their models in realistic settings, fostering research on multimodal attributed graph learning. MM-GRAPH also standardizes GNN architectures, feature encoders, knowledge graph embedding methods, dataloaders and evaluators in order to enable a comprehensive evaluation on multimodal attributed graphs.

To show the importance of these datasets in for advancing multimodal graph learning , we adapt two state-of-the-art multimodal graph neural networks from the recommendation systems to the graph learning domain. These models have demonstrated strong performance in leveraging multimodal information for recommendation tasks. By applying them to the datasets in MM-GRAPH, we aim to provide a comprehensive evaluation of their effectiveness in various graph learning scenarios. We conduct an extensive study on the performance of MMGCN Wei et al. (2019), MGAT Tao et al. (2020), VISTA Lee et al. (2023) and other popular graph learning methods, such as GraphSAGE Hamilton et al. (2017), in the presence of multimodal node features.

Surprisingly, we find that GNNs specifically designed for multimodal scenarios, such as MMGCN and MGAT, do not consistently outperform general-purpose GNN models like GraphSAGE across all datasets in MM-GRAPH. This finding highlights the need for a standard multimodal graph benchmark to rigorously evaluate the performance of different models and rethink where we are in terms of multimodal attributed graph learning. Moreover, while multimodal knowledge graphs (MMKG) exist in the community, there is only a limited amount of datasets available, most of which face issues such as outdated or inaccessible image URLs and poor dataset quality Liu et al. (2019); Safavi & Koutra (2020); Liang et al. (2024); Chen et al. (2024). As a result, higher-quality MMKG datasets are strongly needed.

MM-GRAPH fills this gap by providing a diverse set of datasets with varying scales, tasks, and modality combinations. By establishing a common ground for evaluation, MM-GRAPH enables researchers to reflect the strengths and limitations of existing multimodal graph learning algorithms and identify areas for improvement. Moreover, MM-GRAPH opens up new research opportunities in multimodal graph learning. The availability of datasets with rich textual and visual information allows researchers to explore novel architectures and techniques that can effectively capture and integrate the semantic information from multiple modalities. This can also lead to the development of more advanced and robust graph learning algorithms that can handle the complexities of real-world multimodal graph data.

**Contributions and novelty.** In this work, we introduce a multimodal graph benchmark known as MM-GRAPH. As design principles, we strive to create non-trivial, realistic data that comes from real-world applications and will benefit real-world deployments. Specifically, MM-GRAPH contains

Table 1: Overview of the datasets in the proposed MM-GRAPH. We present five multimodal graph datasets with varying scale and tasks. All datasets are splitted in random. LP = Link Prediction. NC = Node Classification. KGC = Knowledge Graph Completion. OR = Original.

| Name | Text Features | Visual Features | Proposed Task | Metrics | Scale | Split Ratio |
|------|--------------|-----------------|---------------|---------|-------|-------------|
| Amazon-Sports | Product Titles | Product Images | LP | MRR, Hits@K | Small | 8/1/1 |
| Amazon-Cloth | Product Titles | Product Images | LP | MRR, Hits@K | Medium | 8/1/1 |
| Goodreads-LP | Book Description | Book Images | LP | MRR, Hits@K | Large | 6/1/3 |
| Ele-fashion | Fashion Titles | Fashion Images | NC | Accuracy | Medium | 6/1/3 |
| Goodreads-NC | Book Description | Book Images | NC | Accuracy | Large | 6/1/3 |
| MM-CoDEx-s | Entity Description | Entity Images | KGC | MRR, Hits@K | Small | OR |
| MM-CoDEx-m | Entity Description | Entity Images | KGC | MRR, Hits@K | Medium | OR |

three link prediction datasets, two node classification datasets, and two knowledge graph completion datasets as shown in Table 1. We summarize our novel contributions as follows: (1) To the best of our knowledge, no existing standardized graph benchmark provides both text and image features across tasks. This is important as images provide complementary context and details. While one can extract keywords and intent from text, images supply visual cues about objects, scenes, emotions, etc. that give richer understanding, especially for more abstract or complex tasks. (2) We provide a comprehensive analysis on the performance of various methods on MM-GRAPH, highlighting the need of developing new multimodal GNNs. (3) In terms of graph benchmarks, MM-GRAPH is the first benchmark to include not only various GNNs but also different ways of encoding node features, which presents an important and practical question of which feature encoder gives the best features for structure learning tasks.

## 2 RELATED WORK

**Multimodal Feature Learning.** Multimodal feature learning aims to learn joint representations from multiple modalities, such as text, images, and audio. Transformer-based models have shown remarkable success in multimodal feature learning Xu et al. (2023); Radford et al. (2021). These models learn typically transferable visual representations by leveraging corresponding natural language supervision. Models like FLAVA Singh et al. (2022) and Perceiver Jaegle et al. (2021) have demonstrated the effectiveness of jointly pre-training transformers on unpaired images and text while CLIP has shown that contrastive objectives can effectively align representations from different modalities Singh et al. (2022); Jaegle et al. (2021); Radford et al. (2021). We refer to the survey for more details Xu et al. (2023).

**Multimodal Graph Learning.** While most existing multimodal learning approaches focus on 1-to-1 mappings between modalities, real-world data often exhibits more complex many-to-many relationships that can be represented as graphs. Multimodal graph learning (MMGL) aims to address this challenge by leveraging graph structures to learn from multimodal data with complicated relations among multiple multimodal neighbors Ektefaie et al. (2023); Yoon et al. (2023). Recent efforts in MMGL have focused on integrating GNNs/Knowledge Graph Embeddings(KGEs) with pretrained language models (LMs) Yan et al. (2023). Our work builds upon the recent advances in multimodal feature learning and MMGL by introducing a comprehensive benchmark that incorporates both structured and unstructured modalities. We adapt up-to-date multimodal feature learning frameworks and state-of-the-art multimodal graph learning and provide an extensive study on their performance in the presence of features from various modalities.

**Benchmarks for graph representation learning.** Several established graph benchmarks have been developed and widely adopted Morris et al. (2020); Hu et al. (2020); Freitas & Dong (2021). However, when it comes to learning on graphs with rich features, these benchmarks exhibit notable deficiencies. Firstly, these datasets suffer from the absence of raw features, limiting the investigation of attribute modeling. Secondly, these datasets often neglect to explore the impact of feature modeling. Thus, there is a compelling necessity to construct a comprehensive graph benchmark with rich natural features. Yan *et al.* proposed the first standardized graph benchmark with rich textual information, fostering the development of encompassing GNNs with prevalent language models Yan et al. (2023). However, we argue that it is critical to investigate modalities other than text, especially images in

structure learning tasks and proposing the first standarized multimodal graph benchmark to foster the development of research in this direction.

# 3 MM-Graph: The Multimodal Graph Benchmark

In this section, we discuss the proposed 7 real-world datasets in MM-Graph. The basic information is shown in Table 1.

## 3.1 Data Curation

We first introduce the data curation process of all datasets, as well as their license.

**Amazon-Sports** Amazon-Sports is a link prediction dataset, based on the Amazon-Review dataset Ni et al. (2019); Hou et al. (2024), where each node corresponds to a product on Amazon in the sports category and the link captures whether two products are co-purchased together. The text features are the titles of the products and the visual features are the high-resolution raw images of the products. In Figure 1(a), we show an example of nodes and edges in Amazon-Sports. The visual features are images of tents of various shapes under diverse backgrounds and text features are the title of the tents. Specifically, we use the data pre-processing script provided by Patton Jin et al. (2023b).

**Amazon-Cloth** Similar to Amazon-Sports, Amazon-Cloth is constructed according to the co-purchasing information of Amazon products in the clothes category. The text features are titles of products, e.g. Nike mens Revolution 6 Road Running and the visual features are corresponding images. We follow the same data curation procedure.

**Goodreads-LP** is based on the Goodreads Book Graph dataset Wan & McAuley (2018); Wan et al. (2019). Here we construct the graph in the way where each node corresponds to a book on Goodreads and the link captures if a user who likes this book will like the other book, following Zhu et al. (2024a). The text features are the descriptions of the books and the visual features are the cover of the book. Nodes without images are removed.

**Goodreads-NC** is a node classification dataset, based on the Goodreads dataset Wan & McAuley (2018); Wan et al. (2019), where each node corresponds to a book on Goodreads and the link captures if a user who likes this book will like the other book. The text features are the descriptions of the books and the visual features are the cover image of the book. Nodes without images are removed.

**Ele-fashion** is a node classification dataset, based on the Amazon-Fashion dataset Ni et al. (2019); Hou et al. (2024), where each node corresponds to a product and the link captures if a user who buy this product will also buy the other product. The text features are the titles of the products and the visual features are the images of the products.

**MM-CoDEx-s** is a knowledge graph completion dataset, based on the CoDEx-s dataset Safavi & Koutra (2020). The text features are the description of the wikipedia article. For visual features, we crawl the images of the person/place/item from the Wikipedia using Beautiful Soup.

**MM-CoDEx-m** Similar to MM-CoDEx-s, MM-CoDEx-m is based on CoDEx-M dataset Safavi & Koutra (2020). And text features are the description of the wikipedia article. For visual features, we crawl the images from the Wikipedia using Beautiful Soup.

**Data Availability and Ethics.** Our benchmark is organized from existing open source data, with proper open source licenses. Amazon-Sports and Amazon-Cloth and Ele-Fashion are available with Apache License [1]. Goodreads-LP, Goodreads-NC, MM-CODEX-S and MM-CODEX-M are released under MIT License [2] [3]. These do not involve interaction with humans or private data.

---

[1] https://github.com/PeterGriffinJin/Patton

[2] https://mengtingwan.github.io/data/goodreads.html

[3] https://github.com/tsafavi/codex/tree/master

Table 2: Detailed graph-based statistics of datasets proposed in MM-GRAPH. CC = Cluster Coefficient. RA = Resource Allocation. N/A = Not Applicable (Nodes do not have node labels). Detailed definition for each statistics provided in the Appendix.

| Name | Nodes | Edges | Average Degree | Average CC | Average RA | Transitivity | Edge Homophily |
|------|-------|-------|----------------|------------|------------|--------------|----------------|
| Amazon-Sports | 50,250 | 356,202 | 14.18 | 0.4002 | 0.3377 | 0.2658 | N/A |
| Amazon-Cloth | 125,839 | 951,271 | 15.12 | 0.2940 | 0.2588 | 0.1846 | N/A |
| Goodreads-LP | 636,502 | 3,437,017 | 10.79 | 0.1102 | 0.0685 | 0.0348 | N/A |
| Goodreads-NC | 685,294 | 7,235,084 | 21.11 | 0.1614 | 0.1056 | 0.0498 | 0.6667 |
| Ele-fashion | 97,766 | 199,602 | 4.08 | 0.1730 | 0.1467 | 0.0560 | 0.7675 |

Table 3: Detailed MMKG statistics of datasets proposed in MM-GRAPH. (+): Positive (true) triples. (-): Verified negative (false) triples.

| Name | Entities | Relations | Train (+) | Valid (+) | Test (+) | Valid (-) | Test (-) |
|------|----------|-----------|-----------|-----------|----------|-----------|----------|
| MM-CoDEx-s | 1,383 | 39 | 14,298 | 784 | 802 | 1028 | 1074 |
| MM-CoDEx-m | 7,697 | 51 | 47,617 | 2,628 | 2,595 | 4,721 | 4,746 |

## 3.2 TASKS

**Link Prediction.** The task is to predict new association edges given the training edges. The evaluation is based on how well a model ranks positive test edges over negative test edges. Specifically, for Amazon-Sports and Amazon-Cloth, we generate hard valid/test negatives using HeaRT Li et al. (2024a). HeaRT is recognized as a better way of generating negatives for link prediction in the community. For Goodreads-LP, since it takes more than 120 hours to generate hard negatives using HeaRT, we perform random sampling. Each positive edge in the validation/test set is ranked against 150 hard negative edges. For *evaluation metrics*, we report MRR, Hits@10, and Hits@1 - the three most commonly-used evaluation metrics for link prediction Hu et al. (2020); Li et al. (2024a). For Amazon-Sports and Amazon-Cloth, edges are randomly split into train/valid/test splits according to 8/1/1 ratio. For Goodreads-LP, edges are randomly split according to 6/1/3 ratio. Validation and test edges are explicitly removed from the graphs to avoid any potential leakage Zhu et al. (2024b).

**Node Classification.** The task is to predict the book category from $N$ available categories. We report accuracy following Hu et al. (2020); Yan et al. (2023). For Goodreads-NC, there are 10 categories such as History, Children and comics. For Ele-Fashion, there are 12 categories such as shoes, jewelry and dresses. For *evaluation metrics*, we report accuracy, as it's the most common metric for node classification Hu et al. (2020). Nodes are randomly split into train/valid/test following 6/1/3 ratio.

**Knowledge Graph Completion.** The task is to predict missing links in a knowledge graph given entities and relations. Similar to link prediction, the evaluation is based on how well a model ranks positive test edges over negative test edges. Specifically, we use the splits and negatives designed in Safavi & Koutra (2020) to ensure the scope and level of difficulty of the task. Entities without multimodal information (do not have descriptions or images) are filtered out. For *evaluation metrics*, we report MRR, Hits@10, Hits@3 and Hits@1 following Safavi & Koutra (2020).

## 3.3 STATISTICS

We provide graph-related statistics are presented in Table 2 and Table 3. The detailed descriptions for each metric is provided in the Appendix. These statistics provide valuable insights into the structural properties of the graphs in each dataset.

**AMAZON-SPORTS.** There are 50,250 nodes with a total of 356,202 edges in Amazon-Sports, which is a small-scale link prediction dataset. The small scale makes Amazon-Sports great for trying it at first, as it is computation and memory efficient for most applications. The average degree is 14, which indicates that each node have 14 neighbors on average.

**AMAZON-CLOTH.** There are 125,839 nodes with a total of 951,271 edges in Amazon-Cloth. Amazon-Cloth constitutes a medium-scale link prediction dataset. Since the scale of this dataset is much larger than Amazon-Sports, Amazon-Cloth serves as a solid benchmark for real-world e-commerce based

applications. The average degree is 15, which similar to Amazon-Sports. But the average clustering coefficient is much smaller. This indicates that nodes in Amazon-Cloth are more likely to scatter around, instead of forming communities despite its high average degree.

**GOODREADS-LP.** There are 636,502 nodes with a total of 3,437,017 edges, which constitutes a large-scale link prediction dataset. The average degree is 10.

**GOODREADS-NC.** There are 685,294 nodes with a total of 7,235,084 edges, which constitutes a large-scale node classification dataset. The average degree is 21, highest among all datasets.

**ELE-FASHION.** There are 97,766 nodes with a total of 199,602 edges, which constitutes a medium-scale node classification dataset. The average degree is 4, which is the most sparse dataset among MM-GRAPH.

**MM-CODEX-S.** There are 1,373 entities and 39 relations, which constitutes a small-scale dataset.

**MM-CODEX-M.** There are 7,697 entities and 51 relations, which constitutes a medium-scale dataset.

## 4 EVALUATION FRAMEWORK

To enable a comprehensive evaluation on multimodal graph data, MM-GRAPH standardizes the GNN architectures, KGEs, feature encoders, and dataloaders and evaluators used across all datasets.

### 4.1 GRAPH NEURAL NETWORKS

To enable a comprehensive evaluation of GNN architectures on multimodal graph data, MM-GRAPH includes five representative GNN models: GCN Kipf & Welling (2016), SAGE Hamilton et al. (2017), MMGCN Wei et al. (2019), MGAT Tao et al. (2020), and BUDDY Chamberlain et al. (2022) . Additionally, following Hu et al. (2020), we report the performance of an MLP as a baseline to evaluation the usefulness of graph structure.

**Conventional GNNs.** We standardizes GCN, SAGE, MLP for both link prediction and node classification. Since BUDDY is specifically designed for link prediction, so we do not report its performance on node classification.

**Multimodal GNNs.** While GNNs have shown strong performance on tasks involving graphs with unimodal features, there has been limited work on developing GNNs that can effectively handle multimodal graph data. In order to evaluate the effectiveness of GNNs on multimodal graphs, we adapt two recent architectures from the recommendation systems domain to our standardized multimodal graph benchmarks:

MMGCN constructs separate user-item graphs for each modality. The information interchange of users and items in each modality is encoded using GNNs. Modality representations are fused in the prediction layer to predict possible recommendations Wei et al. (2019). For LP, we use the most common dot product decoder to decode possible links. For NC, we stack a 3-layer MLP as transform representations to a number of dimensions equal to the number of node classes.

MGAT extends the popular GAT architecture and learns modality-specific node representations which are then combined using a cross-modal attention layer Tao et al. (2020). This allows MGAT to weigh different modalities based on their importance. Similar to MMGCN, we decode links using dot product for LP and perform NC through a 3-layer MLP.

### 4.2 KNOWLEDGE GRAPH EMBEDDINGS

MM-GRAPH includes two most up-to-date multimodal KGEs: MoSE Zhao et al. (2022) and VISTA Lee et al. (2023). MoSE performs modality split relation embeddings for each modality instead of a single modality-shared one, which alleviates the modality interference. VISTA incorporates the visual and textual representations of entities and relations using entity encoding, relation encoding, and triplet decoding transformers. For VISTA, since the designed architecture scores entities instead of triples, we use the standard 1vsAll setting to report performance instead of hard negatives Lee et al. (2023); Ruffinelli et al. (2020).

Table 4: **Link prediction results on Amazon-Sports, Amazon-Cloth and Goodreads-LP.** Highlighted box indicates the best performing combination for each dataset.

| | Encoder | | Amazon-Sports | | | Amazon-Cloth | | | Goodreads-LP | | |
|---|---|---|---|---|---|---|---|---|---|---|---|
| | Image | Text | MRR↑ | H@1↑ | H@10↑ | MRR↑ | H@1↑ | H@10↑ | MRR↑ | H@1↑ | H@10↑ |
| MMGCN | CLIP | | 31.96 ±0.10 | 16.35 ±0.11 | 68.46 ±0.08 | 22.20 ±0.05 | 10.76 ±0.1 | 46.62 ±0.12 | 31.84 ±0.09 | 18.63 ±0.31 | 59.85 ±0.19 |
| | ViT | T5 | 30.33 ±0.03 | 15.01 ±0.05 | 66.41 ±0.11 | 19.45 ±0.34 | 9.22 ±0.20 | 40.49 ±0.61 | 31.11 ±0.25 | 19.30 ±0.45 | 56.24 ±0.19 |
| | ImageBind | | 31.74 ±0.21 | 16.45 ±0.13 | 67.39 ±0.74 | 24.72 ±0.19 | 12.47 ±0.09 | 51.32 ±0.56 | 26.32 ±0.23 | 16.05 ±0.22 | 46.37 ±0.66 |
| | Dinov2 | T5 | 30.04 ±0.27 | 14.98 ±0.07 | 64.56 ±0.56 | 21.77 ±0.23 | 10.47 ±0.12 | 45.81 ±0.52 | 27.64 ±0.95 | 16.21 ±0.65 | 51.46 ±1.71 |
| MGAT | CLIP | | 27.56 ±0.30 | 13.55 ±0.29 | 60.21 ±0.21 | 21.38 ±0.23 | 10.39 ±0.22 | 44.60 ±0.36 | 74.75 ±1.23 | 64.53 ±1.48 | 92.81 ±0.64 |
| | ViT | T5 | 30.15 ±0.34 | 15.28 ±0.34 | 64.84 ±0.41 | 20.59 ±0.41 | 9.79 ±0.30 | 43.44 ±0.76 | 75.26 ±1.21 | 65.23 ±1.62 | 92.90 ±1.89 |
| | ImageBind | | 30.15 ±0.12 | 15.50 ±0.05 | 64.20 ±0.43 | 22.13 ±0.27 | 10.96 ±0.15 | 45.84 ±0.57 | 74.77 ±0.49 | 64.95 ±0.61 | 92.51 ±0.47 |
| | Dinov2 | T5 | 28.91 ±0.09 | 14.47 ±0.18 | 62.11 ±0.22 | 21.42 ±0.13 | 10.38 ±0.13 | 44.11 ±0.50 | 74.89 ±1.46 | 64.70 ±1.98 | 92.92 ±0.41 |
| GCN | CLIP | | 31.38 ±0.08 | 16.58 ±0.13 | 66.14 ±0.08 | 22.28 ±0.05 | 11.83 ±0.04 | 43.52 ±0.10 | 25.34 ±0.06 | 13.81 ±0.12 | 50.36 ±0.14 |
| | ViT | T5 | 30.83 ±0.07 | 16.31 ±0.08 | 64.76 ±0.15 | 21.60 ±0.05 | 11.37 ±0.03 | 42.29 ±0.14 | 26.50 ±0.10 | 14.86 ±0.08 | 51.54 ±0.14 |
| | ImageBind | | 31.67 ±0.09 | 16.45 ±0.05 | 65.61 ±0.10 | 22.81 ±0.03 | 12.27 ±0.05 | 44.28 ±0.09 | 27.56 ±1.26 | 14.31 ±1.37 | 57.25 ±0.52 |
| | Dinov2 | T5 | 30.42 ±0.02 | 16.02 ±0.03 | 64.02 ±0.06 | 21.19 ±0.08 | 11.09 ±0.06 | 41.46 ±0.16 | 28.21 ±1.12 | 15.11 ±1.06 | 57.94 ±0.95 |
| SAGE | CLIP | | 33.83 ±0.08 | 17.57 ±0.14 | 71.90 ±0.07 | 24.58 ±0.18 | 12.16 ±0.11 | 51.12 ±0.09 | 44.10 ±1.37 | 32.32 ±1.38 | 69.07 ±1.19 |
| | ViT | T5 | 32.01 ±0.10 | 15.94 ±0.17 | 69.84 ±0.21 | 23.11 ±0.05 | 11.10 ±0.04 | 48.89 ±0.09 | 44.79 ±0.18 | 33.11 ±0.21 | 69.43 ±0.18 |
| | ImageBind | | 34.32 ±0.11 | 17.87 ±0.23 | 73.04 ±0.15 | 25.20 ±0.09 | 12.63 ±0.05 | 52.53 ±0.21 | 34.61 ±0.43 | 23.82 ±0.51 | 56.67 ±0.21 |
| | Dinov2 | T5 | 32.20 ±0.12 | 16.19 ±0.2 | 69.98 ±0.32 | 22.98 ±0.01 | 11.12 ±0.04 | 48.28 ±0.11 | 45.61 ±0.22 | 34.01 ±0.27 | 70.01 ±0.11 |
| BUDDY | CLIP | | 31.55 ±0.13 | 15.05 ±0.43 | 70.92 ±0.25 | 23.44 ±0.26 | 11.06 ±0.20 | 51.08 ±0.5 | 43.25 ±0.23 | 31.84 ±0.35 | 67.93 ±0.03 |
| | ViT | T5 | 30.41 ±0.40 | 14.11 ±0.28 | 69.55 ±0.80 | 22.82 ±0.19 | 10.24 ±0.12 | 51.04 ±0.39 | 43.18 ±0.53 | 31.73 ±0.54 | 67.89 ±0.57 |
| | ImageBind | | 33.02 ±0.44 | 17.61 ±0.43 | 69.17 ±0.44 | 24.35 ±0.24 | 12.05 ±0.46 | 51.44 ±0.87 | 41.56 ±0.61 | 29.89 ±0.91 | 67.41 ±0.05 |
| | Dinov2 | T5 | 30.02 ±0.34 | 13.78 ±0.19 | 69.18 ±0.67 | 22.95 ±0.06 | 10.45 ±0.04 | 50.87 ±0.61 | 43.25 ±0.13 | 31.77 ±0.33 | 68.08 ±0.42 |
| MLP | CLIP | | 28.22 ±0.09 | 14.54 ±0.16 | 59.40 ±0.08 | 21.10 ±0.04 | 10.70 ±0.03 | 42.77 ±0.05 | 11.03 ±0.06 | 4.87 ±0.04 | 21.61 ±0.11 |
| | ViT | T5 | 24.81 ±0.05 | 11.63 ±0.05 | 54.78 ±0.04 | 17.65 ±0.06 | 8.14 ±0.04 | 36.77 ±0.06 | 11.10 ±0.17 | 4.84 ±0.15 | 21.94 ±0.24 |
| | ImageBind | | 30.45 ±0.14 | 15.91 ±0.10 | 64.10 ±0.07 | 22.18 ±0.02 | 11.42 ±0.04 | 44.86 ±0.06 | 7.73 ±0.06 | 3.37 ±0.07 | 13.26 ±0.03 |
| | Dinov2 | T5 | 24.81 ±0.16 | 11.62 ±0.18 | 54.97 ±0.22 | 17.53 ±0.11 | 8.07 ±0.09 | 36.53 ±0.26 | 10.28 ±0.04 | 4.49 ±0.05 | 19.86 ±0.03 |

Note that we do not report GNN performance for KGC because it has been shown that GNN design conflates the scoring functions in KGE, leading to a downgrade in performance Li et al. (2023).

## 4.3 FEATURE ENCODERS

**Text encoders.** For text encoders, we select CLIP Radford et al. (2021), T5 Raffel et al. (2020) and ImageBind Girdhar et al. (2023). T5 is selected as the STOA text embedding models, while CLIP and ImageBind are selected so that the output text representations aligns with visual representations.

**Visual encoders.** For visual encoders, we select CLIP Radford et al. (2021), ViT Dosovitskiy et al. (2020), ImageBind Girdhar et al. (2023) and Dinov2 Oquab et al. (2023). ViT and Dinov2 are selected as two different types of visual encoders, while ViT are explicitly trained with supervision, Dinov2 learns robust visual features without supervision. CLIP and ImageBind are selected so that the output text representations aligns with visual representations. Specifically, Imagebind has potentials to extend multimodal graph learning to other modalities, such as audio and video thanks to its ability to bind all modalities into one embedding space. Studies on various feature encoders give insights on design choices of multimodal graph learning, such as (1) how important it is to align multiple modalities into a unified embedding space, (2) if supervised feature encoders work better than unsupervised feature encoders. Note that feature encoders are only used to extract features and are never trained upon.

## 4.4 DATALOADER AND EVALUATORS

MM-GRAPH provides a standardized dataloader and evaluators implementation for all five datasets. The dataloader is built upon PyTorch's DataLoader class and the Deep Graph Library (DGL) . It efficiently handles the creation and transmission of graph samples, allowing for mini-batch training of GNNs. The evaluator is based on OGB's evaluator and performs standardized and reliable evaluation.

The dataloader supports various sampling strategies, such as neighbor sampling and layer-wise sampling, to scale the training process to large graphs. It also enables distributed training using PyTorch's distributed data parallel (DDP) for faster training on multiple GPUs.

By standardizing the GNNs, KGEs, feature encoders, dataloader and evaluators, MM-GRAPH ensures a fair and comprehensive evaluation of graph learning algorithms on multimodal graph data.

## 5 EMPIRICAL ANALYSIS

We conduct a comprehensive evaluation of our proposed benchmark all of the tasks. Our experiments aim to answer the following research questions:

**(i)** What are the performances of MM-GNNs, GNNs and MM-KGEs on MM-GRAPH?
**(ii)** Are multimodal-GNNs better than conventional GNNs on method?
**(iii)** What is the best approach for encoding features? Is alignment between modalities necessary?
**(iv)** How much performance gain can be achieved by using multimodal features compared to unimodal features alone?

Table 5: **Node classification results on Goodreads-NC and Ele-Fashion.**

|  | Image Encoder | Text Encoder | Ele-fashion | Goodreads-NC |
|---|---|---|---|---|
| MMGCN | CLIP | CLIP | 86.10 ±0.50 | **83.29** ±0.20 |
|  | T5 | ViT | 82.39 ±0.30 | 81.85 ±0.22 |
|  | ImageBind | ImageBind | 86.21 ±0.94 | 80.58 ±1.08 |
|  | T5 | Dinov2 | 85.53 ±0.33 | 82.44 ±0.11 |
| MGAT | CLIP | CLIP | 84.66 ±0.29 | 76.48 ±0.59 |
|  | T5 | ViT | 84.01 ±0.08 | 75.43 ±0.76 |
|  | ImageBind | ImageBind | 86.12 ±0.08 | 69.45 ±6.25 |
|  | T5 | Dinov2 | 84.54 ±0.27 | 74.98 ±1.23 |
| GCN | CLIP | CLIP | 79.83 ±0.03 | 81.61 ±0.01 |
|  | T5 | ViT | 79.63 ±0.07 | 81.67 ±0.03 |
|  | ImageBind | ImageBind | 80.35 ±0.02 | 78.91 ±0.04 |
|  | T5 | Dinov2 | 79.37 ±0.04 | 81.71 ±0.03 |
| SAGE | CLIP | CLIP | 87.10 ±0.02 | 83.30 ±0.02 |
|  | T5 | ViT | 84.41 ±0.09 | 83.03 ±0.04 |
|  | ImageBind | ImageBind | **87.71** ±0.13 | 80.39 ±0.21 |
|  | T5 | Dinov2 | 85.31 ±0.09 | 82.99 ±0.08 |
| MLP | CLIP | CLIP | 85.16 ±0.03 | 72.29 ±0.02 |
|  | T5 | ViT | 84.98 ±0.05 | 67.82 ±0.07 |
|  | ImageBind | ImageBind | 88.73 ±0.01 | 58.75 ±0.05 |
|  | T5 | Dinov2 | 84.87 ±0.01 | 68.83 ±0.03 |

Figure 2: **Multimodal GNNs underperforms conventional GNNs.** We compare the best performance of multimodal GNNs (MMGCN/MGAT) and conventional GNNs (SAGE, GCN, BUDDY). Conventional GNNs consistently perform better across datasets, which justifies the importance of building MM-GRAPH and calls for better multimodal GNN designs.

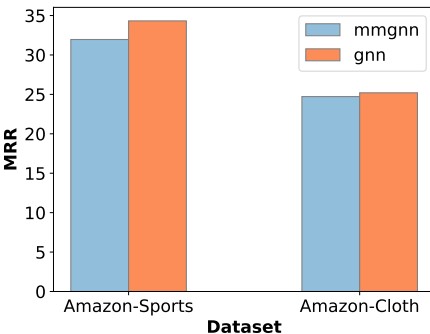

To ensure fair and rigorous comparisons between various feature encoders and GNNs, we adopt consistent experimental settings across all experiments. We perform automatic hyperparameter tuning using Optuna Akiba et al. (2019) to optimize the performance of each model. Detailed experimental setup and hyperparameters are provided in the Appendix C.

### 5.1 RESULTS

The detailed link prediction results on Amazon-Sports, Amazon-Cloth and Goodreads-LP is shown in table 4. The detailed node classification results is shown in table 5. The detailed knowledge graph completion results are shown in table 6. Here are our findings:

**Aligned features demonstrate superior performance, particularly ImageBind.** We evaluate four feature encoders: CLIP and ImageBind, which map features from various modalities to a shared embedding space, and T5ViT and T5Dinov2, which employ state-of-the-art embedding methods for each modality independently without specific alignment layers. Our results reveal performance variations across datasets and tasks.

For *link prediction*, aligned multimodal features consistently outperform unaligned features on Amazon-Sports and Amazon-Cloth datasets, as illustrated in Figure 3. Notably, ImageBind yields the best results across all backbones. On Goodreads-LP, T5Dinov2 exhibits best performance in most cases, suggesting that Dinov2 excels in Optical Character Recognition (OCR) tasks, particularly in understanding book covers.

In *node classification* tasks, CLIP and ImageBind serve as the top-performing feature encoders, further emphasizing the importance of feature alignment.

Similarly, for *knowledge graph completion*, ImageBind consistently demonstrates superior performance across datasets and backbones.

Table 6: **KGC results on MM-Codex-S and MM-Codex-M.** Highlighted box indicates the best performing combination for each dataset.

| KGEs | Dataset | Image Encoder | Text Encoder | MRR | H@1 | H@3 | H@10 |
|------|---------|---------------|--------------|-----|-----|-----|------|
| MoSE | MM-CoDEx-s | CLIP | CLIP | $36.59_{\pm0.16}$ | $28.10_{\pm0.51}$ | $41.79_{\pm1.03}$ | $51.08_{\pm0.29}$ |
| | | T5 | ViT | $39.00_{\pm0.35}$ | $34.40_{\pm0.47}$ | $42.73_{\pm0.33}$ | $46.53_{\pm0.49}$ |
| | | ImageBind | ImageBind | $22.45_{\pm1.36}$ | $5.78_{\pm1.33}$ | $35.48_{\pm1.30}$ | $44.49_{\pm0.98}$ |
| | | T5 | Dinov2 | $40.07_{\pm0.24}$ | $35.82_{\pm1.85}$ | $42.77_{\pm1.17}$ | $47.30_{\pm0.17}$ |
| | MM-CoDEx-m | CLIP | CLIP | $6.97_{\pm0.58}$ | $3.24_{\pm1.11}$ | $9.33_{\pm0.33}$ | $12.15_{\pm0.02}$ |
| | | T5 | ViT | $6.09_{\pm0.01}$ | $1.71_{\pm0.22}$ | $8.42_{\pm0.26}$ | $12.56_{\pm0.28}$ |
| | | ImageBind | ImageBind | $7.01_{\pm0.15}$ | $2.26_{\pm0.33}$ | $9.60_{\pm0.26}$ | $13.98_{\pm0.24}$ |
| | | T5 | Dinov2 | $6.40_{\pm0.18}$ | $2.14_{\pm0.22}$ | $9.06_{\pm0.23}$ | $12.74_{\pm0.14}$ |
| VISTA | MM-CoDEx-s | CLIP | CLIP | $28.49_{\pm0.39}$ | $19.12_{\pm0.64}$ | $30.53_{\pm0.46}$ | $49.83_{\pm0.31}$ |
| | | T5 | ViT | $29.70_{\pm0.51}$ | $20.00_{\pm0.59}$ | $32.11_{\pm0.31}$ | $50.52_{\pm0.75}$ |
| | | ImageBind | ImageBind | $30.39_{\pm0.65}$ | $20.20_{\pm0.84}$ | $33.52_{\pm0.30}$ | $52.68_{\pm0.50}$ |
| | | T5 | Dinov2 | $26.98_{\pm1.68}$ | $17.81_{\pm1.16}$ | $29.53_{\pm2.58}$ | $47.28_{\pm2.86}$ |
| | MM-CoDEx-m | CLIP | CLIP | $22.19_{\pm1.92}$ | $15.81_{\pm1.62}$ | $24.31_{\pm2.10}$ | $34.73_{\pm2.58}$ |
| | | T5 | ViT | $22.10_{\pm1.61}$ | $15.92_{\pm1.33}$ | $24.09_{\pm1.71}$ | $34.68_{\pm2.26}$ |
| | | ImageBind | ImageBind | $23.20_{\pm1.45}$ | $16.97_{\pm1.35}$ | $24.95_{\pm1.24}$ | $35.88_{\pm1.91}$ |
| | | T5 | Dinov2 | $21.38_{\pm1.17}$ | $15.35_{\pm1.15}$ | $23.19_{\pm1.06}$ | $33.40_{\pm1.49}$ |

These findings underscore the significance of aligned multimodal features in various graph-based tasks, with ImageBind showing particularly robust performance across different scenarios.

**Multimodal GNNs perform poorly on the proposed MM-GRAPH datasets.** As shown in Figure 2 and Table 5, we observe that MMGCN and MGAT, the two multimodal GNNs specifically designed to process multimodal input data does not perform better than conventional GNN models such as SAGE across tasks and datasets. As MMGCN and MGAT mostly just perform message passing and aggregation on each modality separately and then fuse the information across modality together at last, we hypothesize that it is likely to do the insufficient fusion of modalities, and calls for better model designs of multimodal structured data.

**Multimodal Features vs. Unimodal Features Only.** To further justify the need and necessity of introducing multimodal graph benchmarks besides the existing text-rich graphs, we further conduct study upon how many add-on can multimodal features bring compared with unimodal text/visual features (T5/ViT). To focus on the effects of features only, we choose MLP as the model and use the best hyperparameters we found using optuna. The results are shown in Fig. 4. Note that we do not report on KGC because all of the KGEs used here require multimodal information as inputs. By using multimodal features, we can consistently get a more than 6 % performance improvement across datasets and tasks. And this justifies the necessity of MM-GRAPH.

## 5.2 CHALLENGES AND FUTURE DIRECTIONS.

First, existing multimodal GNNs are all based on full-batch training schemes and it would go out of GPU memory when the graph is large, e.g. Goodreads-NC and Goodreads-LP. Mini-batch multimodal GNNs are worth exploring in the future. Second, we find that it is hard to find a graph dataset with audio/video features with open-source access, even though music/video recommendation is an task of great importance in real-world deployed systems. We believe the study is similar to what is conducted here and the ImageBind features we use can be easily extended to these scenarios. However, integrating multimodal graphs with audio/video features still require future exploration.

## 6 CONCLUSION

In summary, this paper introduces the Multimodal Graph Benchmark (MM-GRAPH), a comprehensive benchmark for multimodal graph data. MM-GRAPH consists of five diverse datasets spanning various domains and tasks, featuring rich textual and visual node features. To enable a fair and extensive evaluation, MM-GRAPH standardizes the GNN architectures, feature encoders, dataloaders and evaluators used across all datasets. The GNNs include both unimodal and multimodal architectures, while the feature encoders leverage state-of-the-art models for capturing semantic information from text and images. Experiments on MM-GRAPH datasets provide valuable insights into the performance of different GNN and feature encoder combinations in multimodal graph learning settings. By offering a unified evaluation framework, MM-GRAPH aims to facilitate research on

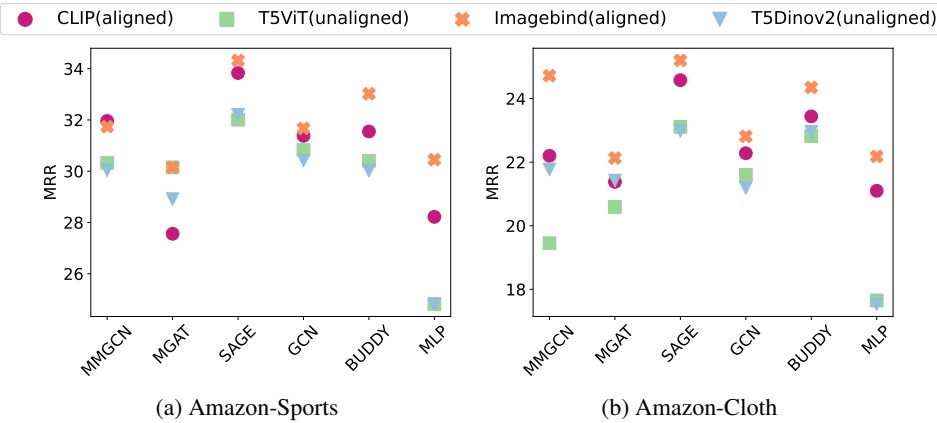

Figure 3: **Feature alignment is important.** We compare the performance of various feature encoders, find aligned features, e.g. CLIP and Imagebind result in much better performance compared with unaligned features on Amazon-Sports and Amazon-Cloth. Among them, Imagebind performs the best across backbones. This indicates the importance of using aligned features on these datasets.

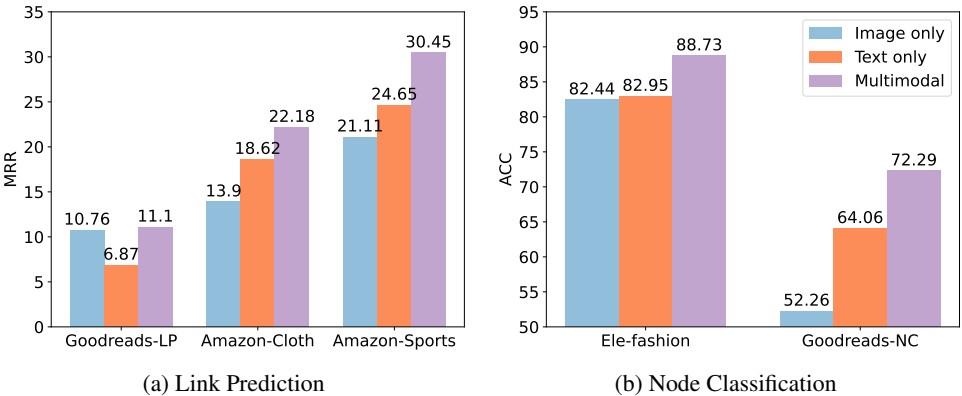

Figure 4: **Multimodal features are helpful for graph learning.** Multimodal features, performs better than text features across datasets and tasks, which justifies the necessity of introducing multimodal graph datasets.

multimodal graph representation learning and drive the development of more powerful and robust graph learning algorithms for real-world applications.

**Ethics Statement.** MM-GRAPH has the potential to significantly advance research on multimodal graph representation learning, which can enable novel applications that positively transform various aspects of society. Many real-world domains, such as healthcare, social networks, and urban computing, involve multimodal graph data. By providing a comprehensive evaluation framework, MM-GRAPH can accelerate the development of more powerful graph learning algorithms for these domains. MM-GRAPH also promotes reproducible research by offering standardized benchmarks, fostering collaboration within the research community. Ultimately, MM-GRAPH aims to unlock the value of multimodal graph data to enable impactful real-world applications. All of the data that MM-GRAPH uses do not involve interaction with humans or private data. However, as with all technologies of this nature, it is important to consider how it can be used responsibly and with responsible development in mind. There are risks of these algorithms perpetuating or amplifying societal biases if the underlying graph data contains biased patterns. Additionally, misuse of multimodal graph can cause disinformation and targeted manipulation. It is therefore crucial that researchers using MM-GRAPH prioritize the responsible development of multimodal graph algorithms in line with ethical principles.

**Reproducibility:** MM-GRAPH code, datasets and details will be publicly available. The datasets will be hosted on Huggingface. The evaluation code and baselines will be hosted on GitHub with documentations.

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
