# Multimodal Attributed Graphs: Benchmarking and Rethinking

## A  Graph Statistics Metrics

To gain insights into the structural properties of the graphs in each dataset, we compute the following graph statistics metrics:

- **Average Degree.** This metric captures the density of the graphs, with higher values indicating denser graphs. It is calculated as the average number of edges connected to each node in the graph.

- **Average Clustering Coefficient (CC).** The clustering coefficient measures the tendency of nodes to form tightly connected groups or clusters. A higher clustering coefficient suggests a greater likelihood of nodes forming densely connected communities. It is computed as the average of the local clustering coefficients of all nodes in the graph.

- **Resource Allocation.** Resource Allocation is a commonly used heuristic for link prediction tasks. It measures the likelihood of two nodes being connected based on their shared neighbors. The performance of this heuristic on each dataset serves as an indication of how well simple heuristic-based methods may perform, providing a baseline for more sophisticated graph learning algorithms.

- **Transitivity.** Transitivity quantifies the probability that two nodes with a common neighbor are also connected to each other, forming a triangle. It is calculated as the ratio of the number of closed triplets (triangles) to the total number of triplets (both open and closed) in the graph.

- **Edge Homophily.** Edge Homophily captures the similarity of node labels with respect to the labels of their neighboring nodes. A high Edge Homophily suggests that nodes with similar labels tend to connect to each other, which can be exploited by graph learning algorithms. Edge Homophily is not applicable (N/A) to link prediction datasets, as nodes in these datasets do not have labels.

## B  Link Prediction Evaluation

**Negative Sampling.** For the task of link prediction, we need to evaluate the performance of models in ranking positive edges (existing edges in the graph) higher than negative edges (non-existent edges). To generate negative edges for evaluation, we adopt the widely-used approach of dedicatedly sampling a fixed number of negative edges (e.g., 150) for each positive edge. This approach has been shown to be more effective than using a single, large set of negative edges for all positive edges **?**.

Given a set of positive edges and their corresponding negative edges, we employ the following evaluation metrics to assess the link prediction performance:

**Hits@K.** This metric measures whether the true positive edge is ranked within the top K predictions made by the link prediction model. Specifically, the model scores and ranks all positive and negative edges. Hits@K is then calculated as the fraction of positive edges that are ranked among the top K predictions.

A higher Hits@K value indicates better performance in ranking positive edges among the top predictions. However, it does not provide information about the specific ranks of positive edges beyond the top K. Therefore, we also report the Mean Reciprocal Rank (MRR) metric.

Table 1: **Best performing hyperparameters under Optuna search on Amazon-Sports, Amazon-Cloth and Goodreads-LP.**

| Encoder | | Amazon-Sports | | Amazon-Cloth | | Goodreads-LP | |
|---|---|---|---|---|---|---|---|
| Image | Text | **LR** | **# Layers** | **LR** | **# Layers** | **LR** | **# Layers** |
| **MMGCN** | | | | | | | |
| CLIP | | 0.002595055 | 1 | 0.001206386 | 2 | 0.001611161 | 3 |
| ViT | T5 | 0.003729380 | 1 | 0.001539690 | 2 | 0.003152644 | 3 |
| ImageBind | | 0.000802559 | 1 | 0.000510544 | 1 | 0.000365131 | 2 |
| Dinov2 | T5 | 0.001184411 | 2 | 0.001575380 | 2 | 0.001036537 | 3 |
| **MGAT** | | | | | | | |
| CLIP | | 0.001706100 | 1 | 0.000109845 | 3 | 0.026710566 | 2 |
| ViT | T5 | 0.001673818 | 3 | 0.000242709 | 3 | 0.016408289 | 3 |
| ImageBind | | 0.00022132 | 2 | 0.000149275 | 2 | 0.000709845 | 2 |
| Dinov2 | T5 | 0.000121652 | 2 | 0.000078832 | 3 | 0.022277487 | 2 |
| **GCN** | | | | | | | |
| CLIP | | 0.001992588 | 1 | 0.001250668 | 1 | 0.000459997 | 1 |
| ViT | T5 | 0.000474838 | 1 | 0.000386748 | 1 | 0.000269508 | 1 |
| ImageBind | | 0.000217399 | 1 | 0.000258416 | 1 | 0.000344080 | 1 |
| Dinov2 | T5 | 0.000386748 | 1 | 0.000504298 | 1 | 0.000579030 | 2 |
| **SAGE** | | | | | | | |
| CLIP | | 0.000888801 | 3 | 0.0006242779 | 3 | 0.001115782 | 3 |
| ViT | T5 | 0.000906908 | 3 | 0.000915933 | 3 | 0.001115782 | 3 |
| ImageBind | | 0.000753606 | 3 | 0.001115782 | 3 | 0.000402602 | 3 |
| Dinov2 | T5 | 0.000915933 | 3 | 0.000915933 | 3 | 0.001115782 | 3 |
| **BUDDY** | | | | | | | |
| CLIP | | 0.000199618 | 1 | 0.000107312 | 2 | 0.000313165 | 1 |
| ViT | T5 | 0.000178234 | 1 | 0.000104734 | 1 | 0.000101783 | 1 |
| ImageBind | | 0.000108298 | 3 | 0.000104882 | 3 | 0.001856957 | 1 |
| Dinov2 | T5 | 0.000269508 | 1 | 0.000138518 | 1 | 0.000199618 | 1 |
| **MLP** | | | | | | | |
| CLIP | | 0.000598264 | 1 | 0.000425976 | 1 | 0.000231730 | 3 |
| ViT | T5 | 0.000276812 | 3 | 0.000386748 | 1 | 0.0001156508 | 3 |
| ImageBind | | 0.000151184 | 2 | 0.000173046 | 2 | 0.000101783 | 3 |
| Dinov2 | T5 | 0.000231730 | 3 | 0.000386748 | 1 | 0.000115650 | 3 |

**Mean Reciprocal Rank (MRR).** MRR is a widely-used metric that considers the specific ranks of positive edges in the predicted ranking. For each positive edge, the reciprocal of its rank in the predicted ranking is calculated. MRR is then computed as the mean of these reciprocal ranks over all positive edges.

A higher MRR value indicates better ranking performance, with a perfect ranking yielding an MRR of 1. MRR provides a more nuanced evaluation of the ranking quality compared to Hits@K, as it considers the entire ranking rather than just the top K predictions.

By reporting both Hits@K and MRR, we provide a comprehensive evaluation of link prediction performance on the multimodal graph datasets in MM-GRAPH. These metrics enable researchers to assess the strengths and weaknesses of different graph learning algorithms in ranking positive edges higher than negative edges, a crucial task in many real-world applications involving graph data.

## C  EXPERIMENTAL DETAILS

To ensure a fair and comprehensive evaluation of different graph learning algorithms on the Multi-modal Graph Benchmark (MM-GRAPH), we conduct extensive experiments with rigorous experimental settings and hyperparameter tuning. In this section, we provide detailed information about the experimental setup, hyperparameter search process, and optimization strategies employed in our study.

### C.1  HYPERPARAMETER TUNING

Proper hyperparameter tuning is crucial for obtaining reliable and meaningful results when evaluating machine learning models. To this end, we perform automatic hyperparameter search using Optuna **?**, a state-of-the-art hyperparameter optimization framework. The hyperparameter search is directly optimized towards maximizing the evaluation metric of interest for each task.

Table 2: **Best performing hyperparameters under Optuna search for node classification and knowledge graph completion.**

| | Encoder | | Ele-fashion | | Goodreads-NC | |
|---|---|---|---|---|---|---|
| | Image | Text | **LR** | **# Layers** | **LR** | **# Layers** |
| MMGCN | CLIP | | 0.01617331603 | 1 | 0.001550788121 | 2 |
| | ViT | T5 | 0.0001215688562 | 2 | 0.0005559463085 | 3 |
| | ImageBind | | 0.01590607512 | 2 | 0.0008154622042 | 3 |
| | Dinov2 | T5 | 0.01233773648 | 1 | 0.0008538306779 | 3 |
| MGAT | CLIP | | 0.0003211703387 | 1 | 0.00004037581919 | 1 |
| | ViT | T5 | 0.0004549756184 | 2 | 0.00001166629883 | 1 |
| | ImageBind | | 0.003739550104 | 2 | 0.00006190162508 | 2 |
| | Dinov2 | T5 | 0.0002344834419 | 1 | 0.00004955270237 | 1 |
| GCN | CLIP | | 0.002747501509 | 2 | 0.0004599971993 | 1 |
| | ViT | T5 | 0.0007123650663 | 2 | 0.0002695086535 | 1 |
| | ImageBind | | 0.0007123650663 | 2 | 0.0003440800285 | 1 |
| | Dinov2 | T5 | 0.0007123650663 | 2 | 0.0005790300826 | 2 |
| SAGE | CLIP | | 0.003235718844 | 3 | 0.001115782634 | 3 |
| | ViT | T5 | 0.001523236953 | 2 | 0.001115782634 | 3 |
| | ImageBind | | 0.00174700527 | 2 | 0.0004026026641 | 3 |
| | Dinov2 | T5 | 0.00134217448 | 2 | 0.001115782634 | 3 |
| MLP | CLIP | | 0.004407317752 | 3 | 0.00174700527 | 3 |
| | ViT | T5 | 0.001342211088 | 2 | 0.0004414532347 | 3 |
| | ImageBind | | 0.00278539483 | 2 | 0.0001409760217 | 3 |
| | Dinov2 | T5 | 0.001287758261 | 3 | 0.0004611586099 | 3 |

Table 3: **Best performing hyperparameters under Optuna search on CoDEx-S and CoDEx-M.**

| KGEs | Dataset | Image Encoder | Text Encoder | **LR** |
|---|---|---|---|---|
| MoSE | MM-Codex-S | CLIP | CLIP | 0.001036811402 |
| | | T5 | ViT | 0.001256901044 |
| | | ImageBind | ImageBind | 0.001016544001 |
| | | T5 | Dinov2 | 0.001004609775 |
| | MM-Codex-M | CLIP | CLIP | 0.001047768667 |
| | | T5 | ViT | 0.001165631895 |
| | | ImageBind | ImageBind | 0.001010501917 |
| | | T5 | Dinov2 | 0.001000862028 |
| VISTA | MM-Codex-S | CLIP | CLIP | 0.001253566266 |
| | | T5 | ViT | 0.001020776224 |
| | | ImageBind | ImageBind | 0.001023083072 |
| | | T5 | Dinov2 | 0.001126222445 |
| | MM-Codex-M | CLIP | CLIP | 0.001003607211 |
| | | T5 | ViT | 0.001011538212 |
| | | ImageBind | ImageBind | 0.001028089197 |
| | | T5 | Dinov2 | 0.001001844192 |

Specifically, for the link prediction task, we optimize the hyperparameters to maximize the Mean Reciprocal Rank (MRR) metric, as it provides a more nuanced evaluation of ranking performance compared to Hits@K. For the node classification task, we optimize the hyperparameters to maximize the Accuracy metric. The hyperparameter search space includes the following key hyperparameters:

- **Learning Rate (LR)**: We explore a wide range of learning rates, from 1e-1 to 1e-5, to find the optimal value for each model and dataset combination.
- **Number of GNN Layers (# Layers)**: We vary the number of Graph Neural Network (GNN) layers from 1 to 3, as the depth of the GNN architecture can significantly impact its performance.

To ensure robust and reliable results, we perform 20 independent hyperparameter studies for each combination of feature encoders and GNN models on a single A40 GPU. Each experiment is run three times with different random seeds, and we report the mean and standard deviation of the respective performance metrics.

## C.2 OPTIMIZATION

All experiments are run with 3 different random seeds to account for the stochasticity in model training and initialization. We report the mean and standard deviation of the respective performance metrics (MRR for link prediction and Accuracy for node classification) across the 3 runs.

For optimization, we use the Adam optimizer with default settings, which has been shown to work well for a wide range of deep learning tasks. To further improve the learning process, we employ a learning rate scheduler with a decay factor of $\gamma = 0.1$ and a step size of 5 epochs. This allows the model to make larger updates in the early stages of training and fine-tune the parameters in the later stages.

The best-performing hyperparameters for each combination of feature encoders and GNNs are presented in Table 1, Table 2 and Table 3. These hyperparameters are selected based on the highest mean performance across the 3 runs.