# OpenReview forum: "Multimodal Attributed Graphs: Benchmarking and Rethinking"
_ICLR.cc/2025/Conference — ICLR 2025 Conference Withdrawn Submission_

### Official Review · Reviewer_G9KR · 2024-10-18

**Soundness:** 3
**Presentation:** 2
**Contribution:** 2
**Rating:** 5
**Confidence:** 4

**Summary:**

This work proposes the multimodal graph benchmark (MM-Graph) which consists of seven datasets over three different graph tasks, providing comprehensive investigation of multimodal graph learning. MM-Graph evaluates different GNN architectures and feature encoders for link prediction, node classification, and knowledge graph completion. The experimental results reveals several meaningful conclusions that are able to facilitate the development of multimodal graph learning field.

**Strengths:**

1. The graph learning tasks include link prediction, node classification, and knowledge graph completion, which is sufficient to comprehensively evaluated the multimodal graph learning methods.
2. The evaluation framework is introduced in detail and easy to understand the overall pipeline.
3. The evaluated methodologies are state-of-the-art, promoting the development of multimodal graph learning field.

**Weaknesses:**

1. Some phenomena within the experimental result are neglect and lack further discussion. First, the authors claim that aligned features are superior, according to the link prediction result in Table 4. This conclusion can be derived from the results of Amazon-Sports and Amazon-Cloth where aligned features achieve best performance in 35 metrics (36 in total). However, for the Goodreads-LP dataset, aligned features only achieve 5 best performance of 18 metrics, losing the predominant superiority. Therefore, the distinction of Goodreads-LP dataset against the two Amazon datasets and the reason of the underperformance of aligned features deserve detailed discussion. Second, for the KGC result in Table 6, the inferiority of aligned features lacks discussion as well. Both T5-ViT and T5-Dinov2 outperform the aligned features significantly, which should not be overlooked. Third, according to the KGC result in Table 6, when switching the KGE method from MoSE to VISTA, one can note that the KGC performance on MM-CoDEx-m is largely improved for all methods. However, different tendency resides in MM-CoDEx-m, where the KGC performance mostly degrades. Hence, the correlation between the KGE method and the dataset scale needs more investigation as well.
2. The difference of multimodal GNNs and the standard GNNs need further elaborations. The manuscript points out that the multimodal GNNs fuse modality information at last, while the strategy of how standard GNNs deal with the multimodal input lacks introduction.
3. The table format is chaos that incurs confusion. First, for Table 1, the alignment strategy is not unified where leftmost, center, and rightmost occur meanwhile. Especially, "Proposed Task" and "Metrics" are nearly overlapped. Second, the best performance in Table 4 is highlighted in the architecture-wise manner, but that in Table 5 is architecture-agnostic.
4. The manuscript organization can be polished up. First, the dataset introduction in Section 3.1 contains too much duplicate sentences. Homologous or similar datasets, e.g., Amazon-Sports and Amazon-Cloth, Goodreads-LP and Goodreads-NC,  MM-CoDEx-s and MM-CoDEx-m, can be introduced together. Second, the statistics in Section 3.3 are redundant, since Table 2 and Table 3 contain the same information. Appending more discussions and analyses on the experiments is recommended.

**Questions:**

1. What is the correlation of the MM-Graph benchmark to the robust graph learning algorithms mentioned in introduction?
2. This work claims that vision information is able to provide richer understanding for abstract or complex tasks. However, the inferiority of Image-only model to the Text-only model may contradict such a judgment.

---

### Official Review · Reviewer_6w8v · 2024-11-03

**Soundness:** 2
**Presentation:** 2
**Contribution:** 2
**Rating:** 3
**Confidence:** 4

**Summary:**

The paper introduces the *Multimodal Graph Benchmark* (MM-GRAPH), a comprehensive evaluation framework for graph learning algorithms on multimodal graphs, combining text and visual features. MM-GRAPH includes seven datasets across tasks like node classification, link prediction, and knowledge graph completion, allowing researchers to assess GNN models in realistic settings rigorously.  MM-GRAPH advances research by standardizing evaluation across datasets, models, and feature encoders.

**Strengths:**

1. **Originality:** This paper fills a gap in graph learning benchmarks by introducing a comprehensive multimodal benchmark that incorporates both text and image features.

2. **Quality:** The benchmark covers various datasets and tasks, such as node classification, link prediction, and knowledge graph completion, providing strong support for research on multimodal graphs.

3. **Clarity:** The paper is well-organized and clearly articulates each dataset curation.

4. **Significance:** MM-GRAPH enables researchers in the graph community to focus more on graph data with rich node attributes in real-world scenarios, helping future graph algorithms become better suited for practical applications.

**Weaknesses:**

1. The paper lacks sufficient references to support key claims, leaving certain arguments, like the added value of visual information for understanding complex tasks, less substantiated. Providing citations or examples from relevant studies could strengthen these assertions.
2. The writing of this paper could be improved. Specifically, the introduction contains multiple similar statements that repeatedly emphasize the potential contributions, which could be more concise. Additionally, the structure of Section 3 is somewhat ineffective, as the dataset statistics are redundantly presented in both text and tables. Moreover, the descriptions in Section 3.1 contain several repetitive phrases about certain datasets, which could be streamlined to make the section more concise overall.
3. The dataset construction in this paper is somewhat flawed. The Goodreads dataset could be used for both node classification and link prediction experiments, so it is not ideal to split it into two separate datasets based solely on task type. The same issue applies to the Amazon dataset.
4. The model selection in this benchmark could be improved. For instance, only T5, CLIP, and ImageBind were considered for text encoders. Recently, many studies have demonstrated that large language models (LLMs) can serve as powerful text encoders, so the authors should consider a more comprehensive approach in the text encoding section.
5. MMGCN and MGAT were originally developed for multimodal recommendation tasks, and it would be helpful for the authors to clarify how these methods can be directly applied to the datasets constructed in this paper. A more detailed explanation of their adaptations or any modifications made for compatibility with the new benchmark datasets would strengthen the understanding of their application in this context.
6. Some issues exist in the presentation of figures and tables in the paper. For example, the text in columns of Tables 1-3 is not centered. In Table 5, the text and image encoders do not align correctly with the models they represent. Additionally, the layout of the bar chart in Figure 4 is visually unappealing and could benefit from improved formatting.
7. The experimental analysis in this paper could be improved. For example, in Table 5, the MLP model performs better than other models on the Ele-fashion dataset but shows a marked decline on the Goodreads-NC dataset. It would be beneficial for the authors to explore and explain the reasons behind this performance discrepancy, such as differences in dataset characteristics or the model’s handling of multimodal features. At the same time, this paper provides a limited analysis of each experimental section and lacks deeper observations and findings.
8. The experimental design section of this paper is lacking, leading to the untrustworthiness of some experimental conclusions. For example, regarding the first conclusion, "Aligned features demonstrate superior performance," the authors did not pair more potential text features with image features. It cannot be ruled out that the superior performance is due to the ImageBind and CLIP text encoders being better than T5, or that their image encoders outperform ViT and Dinov2. The authors should conduct more comprehensive experiments to substantiate this claim.
9. The paper does not explain how the traditional unimodal models, such as GCN, MLP, and SAGE, utilize multimodal features in the experiments. Additionally, I believe that the fusion of multimodal features should be one of the key focuses of this paper.
10. There are some formatting issues in the paper. For example, an extra space appears before the first word in Section 5.2.
11. The paper lacks sufficient innovation. The baselines compared in this work are relatively simple, and the GNN does not incorporate many of the newer models available. Additionally, many works are currently focused on multimodal learning, like MLLM. Evaluating such models on MM-Graph may have more research significance.

**Questions:**

1. I would like to know how the traditional unimodal models in this paper utilize multimodal features.
2. The MM-Graph constructed in this paper seems to have a strong correlation with existing multimodal recommendation systems and multimodal knowledge graphs. I would like to know how the authors view the relationship among these three.
3. Have the authors considered incorporating other modalities, such as audio and video, into the MM-Graph in the future?
4. This paper focuses on tasks like node classification and link prediction. I would like to know if it is appropriate to introduce graph-level tasks in this benchmark.

---

### Official Review · Reviewer_5W3m · 2024-11-04

**Soundness:** 3
**Presentation:** 3
**Contribution:** 2
**Rating:** 5
**Confidence:** 4

**Summary:**

This paper introduces MM-GRAPH, a benchmark for evaluating multimodal attributed graphs that integrate both textual and visual data at the node level. It includes seven datasets across tasks like link prediction and node classification, standardizing GNN architectures, KGEs, feature encoders, dataloaders, and evaluators to enable consistent multimodal graph assessment. The benchmark tests conventional GNNs alongside multimodal GNNs such as MMGCN and MGAT, which separately process different modalities before merging them. Results suggest that multimodal GNNs do not consistently outperform traditional models, highlighting limitations in current fusion strategies. Aligned embeddings (e.g., CLIP, ImageBind) perform better than unaligned ones, underscoring the importance of alignment in multimodal settings. MM-GRAPH thus aims to facilitate more rigorous evaluation and development of multimodal graph learning methods.

**Strengths:**

* MM-GRAPH fills a unique gap by benchmarking multimodal attributed graphs with both textual and visual data, a new addition beyond traditional single-modality benchmarks.
* The benchmark provides a well-structured suite of datasets and a robust evaluation framework, although limited gains in multimodal GNN performance raise questions about model design.
* Explanations are generally clear, though a deeper discussion on multimodal GNN limitations would improve understanding.

**Weaknesses:**

* The image quality in the e-commerce datasets like Amazon-Sports and Amazon-Cloth is lower. Could this impact the process of extracting features? Is there a possibility to improve the quality of these low-resolution images? The current image quality could limit the benchmark’s utility in applications requiring high-resolution visual data, and the potential bias introduced by lower-quality images might reduce the benchmark’s representativeness for high-quality multimodal datasets.
* The benchmark primarily addresses link prediction and node classification, overlooking other common multimodal graph tasks such as recommendation and retrieval, which are often more relevant in real-world applications of multimodal data. This restricted task scope may limit MM-GRAPH’s practical applications, reducing its relevance for research targeting multimodal graph methods in industry-focused domains.
* While the paper shows performance improvements when including visual features, it lacks analysis explaining why visual information specifically improves tasks like node classification. Without a deeper exploration, it remains unclear how images enhance the model’s decision-making. I wonder which aspects of visual information contribute to improved classification, especially in cases where text alone might be sufficient? The authors are recommended to provide an in-depth analysis of the features from each modality that contribute to performance gains. For instance, does visual information help primarily with specific categories or contexts within the node classification task?
* The benchmark does not explore how different domains, such as e-commerce and literature, might exhibit distinct effects based on the integration of text and image features. These differences could be important for interpreting multimodal graph data across various contexts. The lack of domain-specific insights limits its interpretability across diverse data types, which significantly reduce its adaptability for applications in domains with unique modality relationships (e.g., scientific texts with accompanying figures).

**Questions:**

* Has the impact of image quality on model performance been quantified? For instance, do lower-resolution images correlate with lower predictive accuracy in specific tasks?
* Is there a modality-specific analysis that can reveal why multimodal features outperformed unimodal ones in certain datasets?
* Could the benchmark include analyses that reveal how different modality balances affect performance in these distinct contexts?

---

### Official Review · Reviewer_2YLM · 2024-11-04

**Soundness:** 3
**Presentation:** 3
**Contribution:** 3
**Rating:** 5
**Confidence:** 5

**Summary:**

## Summary
This paper constructs a new benchmark named MM-Graph. The authors claim that it is the first comprehensive multi-modal graph benchmark, which incorporates both textual and visual information. It contains 7 graph datasets at various scales. The authors provide extensive studies on performance of various graph learning methods. The datasets and experiments lead to the improved performance of real-world applications on multimodal attributed graphs.

**Strengths:**

## Strength
1. The topic is interesting and practical. And the multi-modal graph datasets are necessary for real-world applications.
2. The experiments and analyses are extensive.
3. The motivation is clear, and the data construction process is easy to understand.

**Weaknesses:**

## Weakness
1. The dataset and codes are not available on anonymous GitHub, limiting the reproducibility.
2. Missing important unsupervised tasks on graphs, i.e., node clustering [1] and graph-level clustering [2]. And missing a comprehensive survey on these topics [1-3].
3. The scalability of the datasets is limited. Although it introduces some middle size datasets like Goodreads-NC, Goodreads-LP, they merely contain 0.6M nodes. Therefore, I think they are only middle-size datasets, especially compared with the exiting large graph benchmarks such as ogbn-papers100M, which contains ~100M nodes.
4. The analyses are good. But they are not very relevant to the proposed benchmark. I recommend conducting more analyses on the proposed benchmark, such as data distribution, stability, scalability as I mentioned above, noise, etc.
5. What’s the difference/features between the proposed benchmark with the existing multi-modal knowledge graph datasets/multi-modal graph datasets?

[1] Dink-Net: Neural Clustering on Large Graphs

[2] GLCC: A General Framework for Graph-Level Clustering

[3] A Survey of Deep Graph Clustering: Taxonomy, Challenge, Application, and Open Resource

**Questions:**

See weaknesses.

---

### Note · Authors · 2024-11-12

I have read and agree with the venue's withdrawal policy on behalf of myself and my co-authors.